# Paving Paths to 2050: Mapping the Mexican Power Sector's Potential to Build Sustainable Futures

Manuel Martínez , Juan Carlos Castro, Carlos David Leal-Fulgencio, Santiago Álvarez-Herrero and Karla Graciela Cedano-Villavicencio *

Instituto de Energías Renovables (IER), Universidad Nacional Autónoma de México (UNAM), Temixco 04510, Mexico; mmf@ier.unam.mx (M.M.); jccad@ier.unam.mx (J.C.C.); cdlef@ier.unam.mx (C.D.L.-F.); sah@ier.unam.mx (S.Á.-H.)
* Correspondence: kcedano@ier.unam.mx; Tel.: +52-7773620090

**Abstract:** A prospective analysis of the Mexican power system has been carried out, and the possibilities for complying with the National Determined Contribution (NDC) commitments adopted in the Paris Agreement in the National Climate Change Strategy of the General Law for Climate Change are presented. A hybrid approach was used, inspired by Foresight Frameworks on one side, and using an accounting and scenario-based modeling platform called Low Emissions Analysis Platform (LEAP) with economic optimization on the other, to analyze four possible scenarios to 2050: Trend, also known as Business As Usual (BAU), which only looks at the growth rate of the gross domestic product (GDP) and population, along with the country's proven potential for renewable resources under the policy framework before 2015; Market, which considers the trend of installation of renewable technologies and the policies of the past administration; Sovereignty, which does the same as the previous scenario but for the policies of the current Mexican government; and Sustainable, which together with GDP and population growth takes into account the possible potential of national renewable resources and characteristics that the country should have to foster this. The research question was: Is there a long-term energy policy for the Mexican power sector to accomplish its NDCs? The study concluded that the best scenario is Sustainable, showing that the possible potential of the country's renewable resources has the capacity to meet the Mexican commitments adopted in the Paris Agreement if, on one side, techno-economic renewable technologies and electricity storage systems are used, and on the other, if the country's strategic areas evolve into a more holistically sustainable future.

**Keywords:** multidisciplinary prospective study; scenarios analysis; energy mix; $CO_2$e emissions; renewable energy



## 1. Introduction

In the framework of the 21st Conference of the Parties (COP 21), held in Paris in 2015, several countries pledged to significantly reduce their greenhouse gas (GHG) emissions to a level that allows the planet's temperature not to rise more than 2 °C. Every 5 years thereafter, all parties must submit their Intended Nationally Determined Contribution (INDC) to the United Nations Framework Convention on Climate Change (UNFCCC). INDCs are a commitment by the international community to reduce national emissions and adapt to the effects of climate change; the parties must adopt internal mitigation measures to achieve the objectives [1].

The United Nations 2030 Agenda for Sustainable Development and its Sustainable Development Goals (SDGs) call on countries to adopt urgent measures to combat climate change and its effects. Guaranteeing access to affordable, safe, sustainable, and modern energy is SDG 7, which is essential for eradicating poverty, protecting the planet and ensuring prosperity. In addition, it constitutes an axis of efforts to face climate change [2].

Responsible energy consumption is one of the indicators of sustainable development most promoted by any country; the need to satisfy this predicted energy demand for a given period is the main challenge for energy planning [3]. Energy planning elaborates and confirms scenarios in the energy economy based on the definition of the World Energy Council that establishes that this "part of the economy related to energy problems, taking into account the analysis of energy supply and demand, as well as the implementation of the means to ensure coverage of energy needs in the national or international context" [4].

Researchers around the world have used different energy models to address energy, economic and environmental policy, and planning concerns. The Low Emissions Analysis Platform (LEAP version 2020.1.106) [5] bottom-up accounting model, applied to the electricity sector, allows for evaluating different energy policies in electricity generation (such as efficient use of energy, fuel substitution, and technological changes) and their corresponding emissions [6]. In 2003, an economic evaluation for the Mexican electricity sector was reported in terms of a cost–benefit analysis on future energy scenarios for Mexico until the year 2025, using the LEAP model [7]. In 2011, Hoseok Kim et al., described the assembly of a model to assess energy futures in the Republic of Korea using the LEAP model and described the results of several policy-based scenarios focusing on different levels of nuclear energy use and their impacts on the energy supply and demand in the Republic of Korea [8]. In that same year, Yanjia Wang et al. [9] reported on the latest development of energy production, energy consumption and strategic energy planning and policies in China, using the LEAP tool for modeling the impacts of the implementation of new and expected energy and environmental policies. In 2014, Nicolas Di Sbroiavacca et al. [10] applied the LEAP models, the Integrated Assessment Model of the Energy Research Centre of the Netherlands (TIAM-ECN), and the Global Change Assessment Model (GCAM) to assess the impact of a variety of climate change control policies in primary energy consumption, final energy consumption, and electricity; the development of the electricity sector; and savings in $CO_2$ emissions from the energy sector in Argentina during the period 2010–2050. In 2017, Nnaemeka Vincent Emodi et al. [3] applied a scenario-based analysis to explore Nigeria's demand, supply and associated GHG emissions in the future from 2010 to 2040 using the LEAP model. Recently, Aristizabal et al. [11] analyzed energy demand and greenhouse gas emissions produced in Colombia using LEAP software, by building a model based on 2015 and with two future scenarios (positive and negative).

As part of the national efforts to regulate greenhouse gas and compound emissions in Mexico, in 2012, the General Law of Climate Change (LGCC from the Spanish acronym, Ley General de Cambio Climático) [12] entered into force, with the objective of establishing the bases for Mexico to contribute to the fulfilment of the Paris Agreement. Its function is to regulate, promote and incorporate adapting and mitigating actions with a long-term approach, define the obligations of the authorities, and establish institutional mechanisms to face the challenge.

To strengthen the LGCC, it is essential to design and implement public policy instruments. One of the planning instruments is the National Climate Change Strategy (ENCC from the Spanish acronym, Estrategia Nacional de Cambio Climático) published in 2013 [13], which describes the lines of action to follow based on the information available on the current environment and future, to address national priorities and achieve the country's long-term goals. The ENCC sets viable goals to combat this phenomenon in the next 40 years; it mentions that the country will grow in a sustainable way and the sustainable management of natural resources, and the use of clean and renewable energies will be promoted. It is intended to mitigate 50% in 2050 in relation to the emissions of the year 2000.

Mexico recognizes that it is important to carry out actions that contribute to the efforts of the international community in GHG mitigation. According to data from the National Inventory of Greenhouse Gases and Compounds (INEGYCEI from the Spanish acronym, Inventario Nacional de Emisiones de Gases y Compuestos de Efecto Invernadero), in 2016, the energy sector was responsible for 72.3% of total national GHG emissions [14]. In this

context, any effort to reduce emissions and mitigate climate change must include this sector. The INEGYCEI is a solid tool. Mexico used this tool and was the first country in Latin America to present its INDCs before the UNFCCC.

In this paper, a study has been conducted regarding the possibilities that the Mexican power sector has in order to comply with the commitments adopted within the international community towards the mitigation of climate change.

A hybrid methodology was used to obtain both a quantitative projection of $CO_2$e emissions, to observe the compliance of Mexican energy policies with the goals set for the Paris Agreement, and a qualitative approach to obtain the possible economic, social, environmental and institutional strategies needed to achieve these goals. So, the first approach used an accounting and scenario-based modeling platform called the Low Emissions Analysis Platform (LEAP), with economic optimization, and a prospective method was utilized to analyze four possible scenarios to 2050, inspired by Foresight Frameworks, considering the strategies proposed by the previous administration of 2012–2018 and the current one of 2018–2024, in conjunction with a trend scenario and a scenario that has no limitations regarding the installation of renewable technology.

*Mexican Electricity Sector*

In 2013, the Energy Reform was approved as part of the efforts to modernize and diversify the Mexican economy [15]. The general objective of this reform was to provide a more sustainable, efficient, transparent, and productive energy sector, to increase the benefits obtained from the country's resources, while promoting the growth of low-carbon energy sources [16]. The National Electric System (SEN from the Spanish acronym, Sistema Eléctrico Nacional) is a component of the Mexican energy sector that involves the generation, transmission, distribution, and commercialization of electrical energy. Under the current regulatory framework, generation technologies are divided into conventional and clean. The group of conventional technologies is made up of plants that generate electricity from fossil fuels as primary energy and do not have $CO_2$ capture and storage equipment. According to the National Inventory of Greenhouse Gases and Compounds Emissions, these types of technologies contribute 18% of total GHG emissions nationwide [17].

Despite the energy subsidy implemented by the Mexican government, which represents 0.75% of the gross domestic product (PIB, in Spanish [18]), the price of electricity is high and is not competitive. Average electricity rates are 25% higher compared to the United States, even with the subsidy; without it, they are 73% higher [19]; this is due to the high prices of fossil fuels that are frequently used in Mexico to generate electricity. While countries such as Colombia, Bolivia, Vietnam, and Indonesia, among others, have made progress in reducing or eliminating the subsidy for electricity users, Mexico maintains this scheme. The electricity subsidy in Mexico increased almost three times in the period from 2014 to 2018, representing USD 14 billion in the last year [20].

The amount used to subsidize conventional electricity could be redirected to boost the use of renewable energy since its potential in Mexico for electricity generation is high. According to the International Renewable Energy Agency (IRENA) ranking in 2019 [21], Mexico was the fourth country in Latin America with the highest wind potential, highlighting states such as Oaxaca, Tamaulipas, Coahuila and Yucatán with air currents of up to 10 m/s (when the world average is about 6.5 m/s). In addition, Mexico is located in the solar belt, with an average solar irradiation higher than the global one, with more than 5 kWh/m$^2$ per day. The areas with the greatest potential are Chihuahua, Durango, Sonora and Baja California, even exceeding 8 kWh/m$^2$ per day [22]. Therefore, Mexico has a potential that is superior to the leaders in photovoltaic power generation, such as Spain or Germany.

The total installed electricity capacity was 78,447 MW in 2019, while in 2020, it increased to 86,034 MW; this increase is mainly due to combined cycle (3344 MW), wind (927 MW) and photovoltaic (2149 MW) power plants. In 2019, the installed capacity of renewable energy plants, such as wind, photovoltaic and bioenergy plants, represented

12.8% and was 10,071 MW, and in 2020, it was 13,180 MW—an increase of 30.87% compared to 2019, with the wind and photovoltaic plants being the main sources of this increase. Regarding the total installed energy capacity by modality in 2020, the National Power Utility (CFE, Comision Federal de Electricidad, in Spanish) is dominating the market with 51.36%, followed by 28.17% for the private sector and, later, 19.39% for independent producers of electric energy (PIE, in Spanish). On the other hand, the production of electrical energy, considering the net generation of the CFE and the different permit holders, during 2019 was 317,820 GWh, of which 25,743 GWh came from renewable energies (8.1%): wind, photovoltaic and bioenergy. While the production of electrical energy in 2020 was 236,628 GWh, of which 24,372 GWh came from renewable energies (10.3%), the highest growth was experienced by photovoltaic energy, which increased by 38% compared to 2019. Conventional internal combustion energies dominated, representing 54.1% for 2019 and 58.2% for 2020 [17].

Mexico has demonstrated its commitment and entrepreneurship in facing climate change. One of Mexico's unconditional goals was to reduce GHG emissions by 22% by 2030. Specifically, the electric power generation sector could contribute with a reduction of 63 MtCO$_2$e, that is, 31% of its emissions [14]. In addition, in the national context, the reforms to the General Law on Climate Change establish that the reduction of greenhouse gas emissions by 2030 may be increased by up to 36%; it is subject to the adoption of an agreement on global issues.

## 2. LEAP Modeling

An accounting and scenario-based modeling platform called the Low Emissions Analysis Platform (LEAP) is used to construct a model of different scenarios of the Mexican power sector. LEAP supports a wide range of different modeling methodologies: on the demand side, these range from bottom-up, end-use accounting techniques to top-down macroeconomic modeling. On the supply side, LEAP provides a range of accounting, simulation and optimization methodologies that are powerful enough for modeling electric sector generation and capacity expansion planning, which are also sufficiently flexible and transparent to allow LEAP to easily incorporate data and results from other more specialized models [23].

### 2.1. LEAP Methodology

The model considers a period from 2017 to 2050. Public official information from 2014 to 2017 was used for its calibration, contemplated in the LEAP methodology as Current Accounts. Also, the generation capacity expansion is adjusted to meet the established operating reserve margin of 21% according to the National Development Program of the Power Sector (PRODESEN, in Spanish, 2018 [17]).

In this model, four scenarios were created: (a) Trend, which reflects the inertial conditions of events that occurred from 2014 to 2017 (as if there were no changes); (b) Sovereignty, which considers the average growth rates projected by the current federal administration for renewable energy technologies [24]; (c) Market, which takes into account the average growth rates obtained from the previous administration, as given by PRODESEN 2018 [17] for renewable technologies; and (d) Sustainable, which has no restrictions on the use of the potential of renewable energies and optimizes the system at the minimum annual system cost, taking into account investment costs, fixed and variable O&M costs, and fuel costs.

In the model, the demand for electricity is discretized by the economic sector: Households, Agriculture, Industry, Commerce, Services and Transport. For each one, their electrical energy consumption [25], their participation in GDP [26], and the national population were considered. In addition, the technical losses of electrical energy in the transmission and distribution processes were included [17].

The demand projection for the prospective period was carried out with the growth rates reported by the International Energy Agency (IEA) [16], the Ministry of Energy

(SENER) [17] and the National Population Council (CONAPO) [27] for these macroeconomic and demographic variables (see Table 1).

**Table 1.** Annual growth rates for population and GDP.

| Variable | Average Annual Growth Rate (Tmca) |
|---|---|
| Population | 0.86% [27] |
| Gross domestic product | 3.0% [16,17,24] |

The electricity demand for all four scenarios will increase at an average annual rate of 2.82%, with a total of 602.54 TWh at the end of the period, as shown in Figure 1.

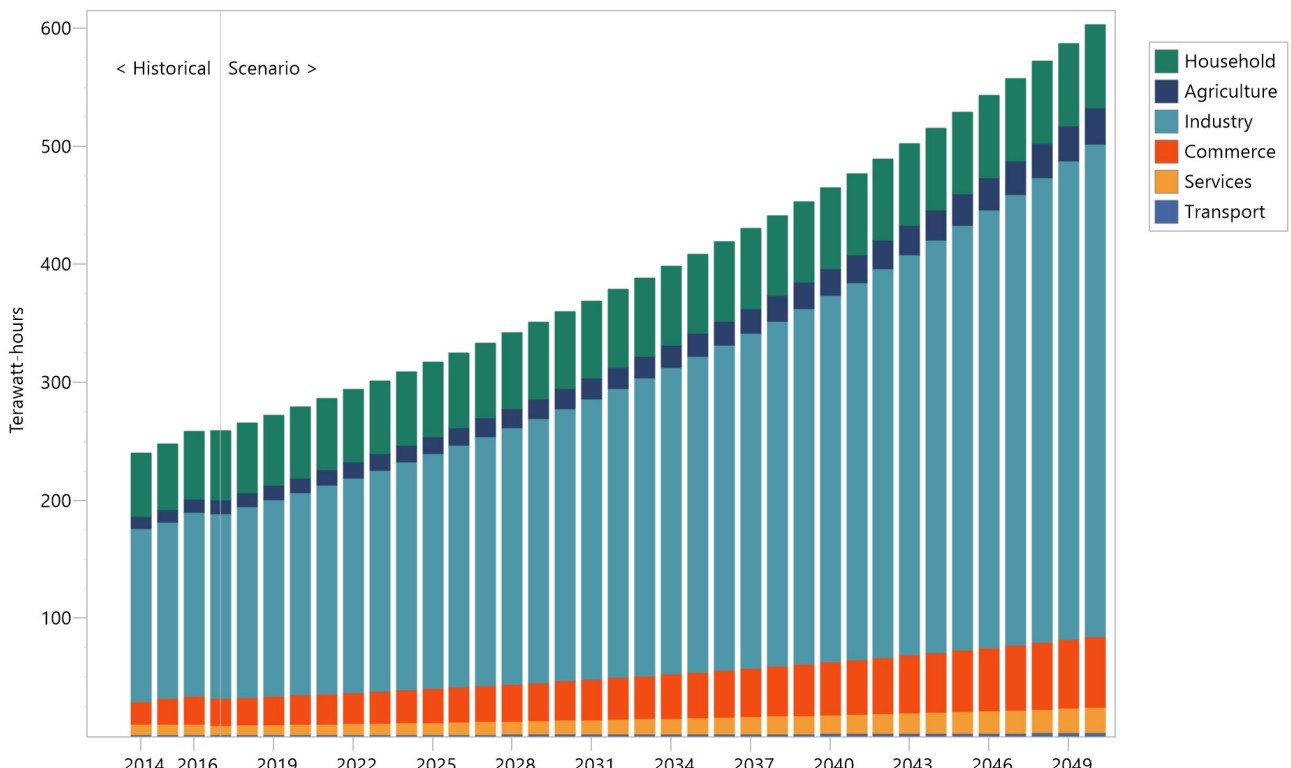

**Figure 1.** National demand for electricity by sector.

Regarding the supply of energy, that is, how the demand is satisfied, 12 technologies for the generation of electricity were considered: Bioenergy, Solar PV, Geothermal, Wind, Hydro, Combined cycle, Fluidized bed, Internal combustion, Gas turbine, Thermoelectric, Nuclear and Coal. Additionally, electricity storage systems were incorporated to expand the PV systems offering. Official reports issued by SENER [28,29] were used to obtain information on each of the technologies, such as: installed capacity, electricity generation, efficiency, lifetime, plant factor, investment costs, fixed and variable operating and maintenance costs, and the potential of renewable energies due to resource availability and proven national reserves of fossil fuels.

For the projection in the prospective period, the corresponding annual growth rates in investment costs and fixed and variable O&M costs of each technology, proposed by IRENA [30] and NREL [31], were considered, together with the annual growth rates of fuel prices from the study Costs and Reference Parameters for the Formulation of Investment Projects in the Power Sector (COPAR) 2015, CFE [32].

Everything described up to this point about the energy supply is what the proposed scenarios have in common. Each of them is described in more detail in the next section.

Finally, to make a comparative analysis between scenarios and determine which is the most favorable for achieving sustainable development with a low impact on climate change, the level of $CO_2e$ emissions is used as a criterion; this is calculated with the model in LEAP and compared with the Baseline and the emission targets presented at the ENCC [12]. If a scenario does not meet these goals, it is not a desirable scenario.

### 2.2. LEAP Scenario Development

Regarding the scenario analysis, four different scenarios have been proposed: Trend, Market, Sovereignty, and Sustainable. Each of them is detailed below.

BAU: This is a trend scenario (Business As Usual), which only contemplates the growth of macroeconomic and demographic variables, as well as the annual growth rates of the investment costs of technologies and fuels. In this scenario, the proven potential of renewable resources is considered, which is defined by SENER in its report "Renewable Energy Prospective 2018–2032" as "the potential that has technical and economic studies that verify the feasibility of its use" [28].

The projection of the installation of technologies for the use of renewable energies of the past administration is very different from that of the current administration. This main difference is what distinguishes the following two scenarios; they are intended to explore the possibilities that one and the other offer with respect to GHG emission targets.

Market: This scenario is based on the projections for the installation of technologies for the use of renewable energies presented at PRODESEN 2018–2032 [17], which was the last one issued by SENER in the previous administration. The government at that time had a policy more open to the participation of private initiatives in the national electricity sector, which promoted the use of renewable energies.

Sovereignty: This scenario considers the trend of the installation of technologies for the use of renewable energies presented in the PRODESEN 2020–2034 [24] issued by SENER in the current administration, which has defined its strategy as "sovereign", by betting on technologies that take advantage of the reserves of fossil sources that the country has.

Sustainable: In this scenario, the investment cost projections and fixed O&M costs are considered according to the report "Renewable Power Generation Costs" [30] for renewable technologies and the projections of the costs of fossil technologies until 2050 according to the 2020 Annual Technology Baseline (ATB) report [31]. In addition, unlike the Trend, it considers the possible potential of renewable resources, defined as: the theoretical potential of the resources but lacking the necessary studies to evaluate the technical feasibility and possible economic, environmental, and social impacts.

### 2.3. LEAP Results

For the resolution of the model, LEAP considered only cost optimization, proposing in each scenario the lowest-cost solution to satisfy the demand for electricity with the different generation technologies considered.

In the BAU scenario, the combined cycle appears as the predominant form of generation; for this part, renewable energies do not have a considerable increase, as shown in Figure 2. It is important to consider how dangerous it is to depend on a single source of energy, especially natural gas, which is used in combined cycle plants, and much of its supply is imported. For example, in February of this year [29], in Mexico, there was an energy crisis in which a large part of the country's population suffered from power outages due to the climate crisis in Texas, US.

From this scenario, one can observe something that is repeated in all other cases: technologies such as Coal-fired, Gas turbine, and even Hydro stop being installed and disappear as soon as the existing plants reach their lifespan.

In the Market scenario, as mentioned above, the growth rate of the installed power of renewable energies was considered in the previous administration, when the generation of electricity was opened to private initiative; that is why it can clearly be seen how the installation of renewable technologies gains ground on those that use fossil fuels, mainly

the Combined cycle (see Figure 3). Solar PV, Geothermal and Wind have considerable growth over time compared to the two previous scenarios, and even the Hydro stops disappearing, as this technology is re-powered and fulfils its lifetime.

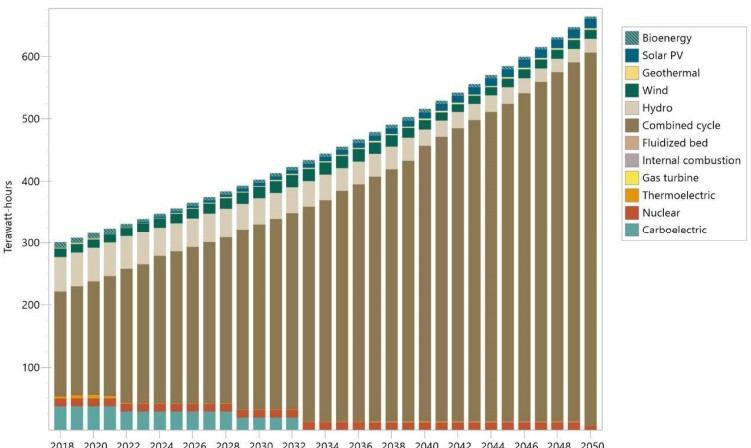

**Figure 2.** Electricity supply in Trend scenario.

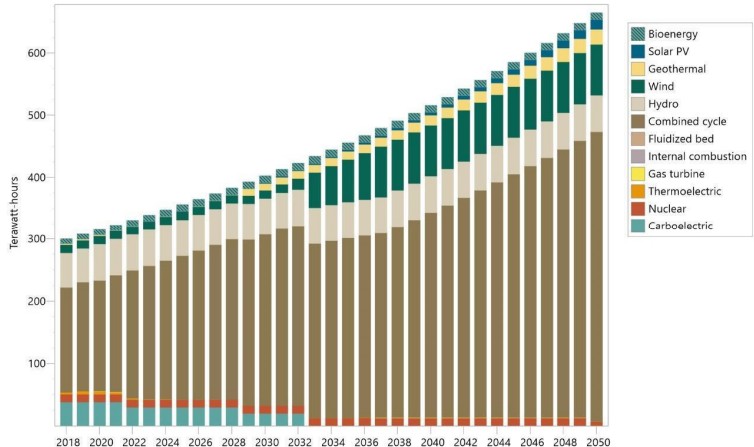

**Figure 3.** Electricity supply in the Market scenario.

In the Sovereignty scenario, a very similar proposal to that of the Trend is seen, since the Combined cycle is also the predominant form of generation; although, on the other hand, there is a growth in geothermal energy that had not appeared in the Trend (see Figure 4). This growth is very small and is because the growth rate in the installed capacity of renewable technologies has been very low in the current administration. We can even see that the plants that reached their lifetime are replaced by a Combined cycle.

Finally, the Sustainable scenario, which considers the country's possible renewable resource, presents a generation proposal with the participation of renewable energies much broader than all the previous ones, with the incorporation of electricity storage systems, as its capital and maintenance costs are expected to decrease and the utility operators facilitate their incorporation to the grid. It is even noticeable how by 2050, renewable energies represent more than 50% of the installed capacity for the generation of electrical energy (see Figure 5). It is worth mentioning that the participation of Geothermal, Wind and Hydro is almost constant from 2040 to 2050; this is due to techno-economic limitations, as well as limits on the availability of potential resources, in comparison with Solar PV. Also, in addition to the environmental aspect, there is a more varied generation set, which does not depend on only one energy source; it would be a mistake not to place importance on the energy security that this represents.

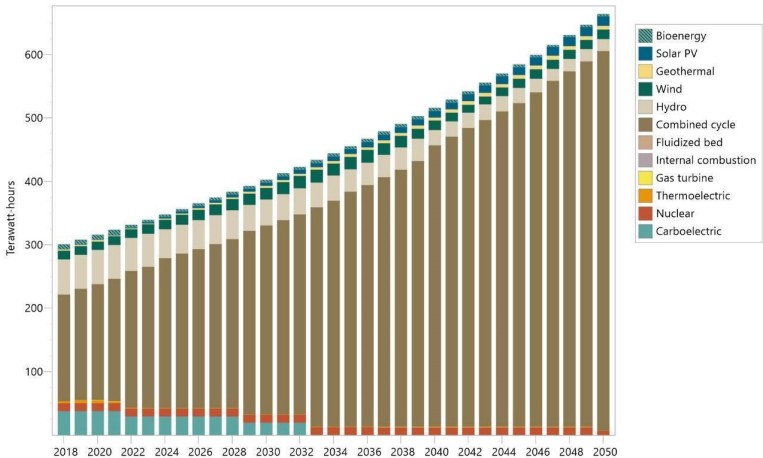

**Figure 4.** Electricity supply in Sovereignty scenario.

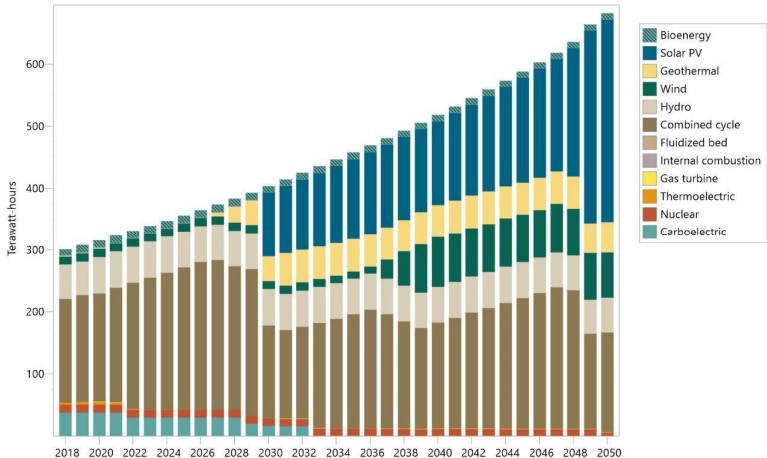

**Figure 5.** Electricity supply in Sustainable scenario.

As mentioned in the methodology, the comparison of scenarios is made in relation to the Baseline and the goals set in the ENCC [18], represented by the red and blue dotted lines of the graph in Figure 6, respectively. The interpretation is simple: if the $CO_2$e emissions of a scenario are above the blue line, then that scenario is not desirable.

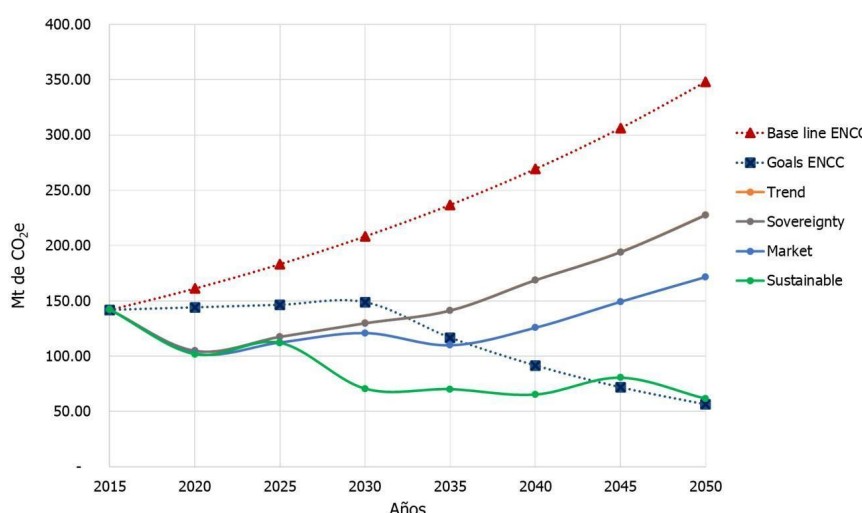

**Figure 6.** GHG emission comparison among scenarios. Trend and Sovereignty Scenarios are overlapped.

The model showed that the emissions of the Trend and Sovereignty scenarios are practically the same, as shown in Figure 6, where they appear superimposed. This could be predicted from the generation proposal of both. Both the Trend and the Sovereignty base practically the entire generation on the Combined cycle; however, in the second, there is a little more installation of renewable technologies, but it is not enough to impact $CO_2e$ emissions. In any case, neither the Trend nor the Sovereignty meet the emissions targets proposed by the ENCC from 2033 onwards. Of the proposed scenarios, these two are the least desirable.

For its part, the Market scenario, which had a proposal with a larger presence of renewable technology than the previous two, has a level of $CO_2e$ emissions significantly lower than the two mentioned scenarios; despite this, and since the emission targets drop considerably from the year 2030, in 2037, this scenario is no longer able to fulfil what is desired.

With the analysis carried out up to this point, if only the proven renewable resources (Trend) are considered, with the annual installation rate of the previous administration (Market) or the current one (Sovereignty), the National Electric System (SEN) would not be able to meet the agreed emission targets for compliance with the Paris Agreement. The Sustainable scenario, which is the one that represented the proposal with the most use of renewable resources of the four, considering the possible resources of the country, perfectly meets the $CO_2e$ emissions goals until 2047, where it barely exceeds the level established by the ENCC, and it remains in this way until the end of the period analyzed.

It should be stressed that up to this point, only an economic optimization has been made and there have been no minimum annual installation restrictions or any other type of conditioning that encourages renewable technologies over others; in other words, if a minimal effort were made in 2047 to have more installed capacity for the use of renewable energies, it would be enough for the SEN to meet the $CO_2e$ emissions targets.

### 3. Futures Studies

Futures studies allow us, based on retrospective analysis and the current situation from defined fields of activities and vectors, to first identify the trending and desirable futures and then, by incorporating links with the external context, recognize the elements that bear the future and propose specific courses of action, as well as establish the possible futures of any institution within social, economic, political and environmental contexts. The institutions to study can be the world, regions, countries, states, universities or companies. Of the various methods that we have used to carry out these studies, on this occasion, we selected the scenario method, a methodology that narrates solid descriptions of the events of an institution in the long term.

Now, since its inception and responding to specific concerns in different countries and historical moments, specific variants have emerged for the study of the future. Among many others, it is worth highlighting: long-range planning, futures research, technological forecasting, futurology, scenarios planning, strategic foresight and human and social foresight. Each of these approaches has a different vocation, specific objectives and specific areas of application, but they all share the same sense and concern for the future.

Thus, these studies are presented as a field of knowledge for the systematic and organized questioning of the future, progressively becoming an academic discipline that deals with long-term studies and the planning instruments that must accompany them [33]. They are focused on conceiving possible, probable, and desired visions of the future [34] and are decision-oriented—that is, they seek to identify and describe the forces and factors that must be understood in the present to carry out intelligent decision-making. It is worth emphasizing that the purpose of these types of studies is not the prediction of specific events; their value should not be considered based on the accuracy or completeness of their descriptions of the following years but on their usefulness for planning and openness to possibilities and change [35].

### 3.1. Methodology

To build scenarios, we have developed a methodology inspired by that described by Hines and Bishop [36], known as Foresight Frameworks. This methodology seeks to focus on the most important information. It helps to prevent the product of quantitative analysis from being seen simply as a mass of information.

The method offers value to the foresight community as a training tool and as an integrative practice. Users have found it useful for identifying and analyzing the information required to carry out a long-term planning project and organize it into a logical flow. We have two vectors to conduct the analysis: that of the key future drivers, which are those that are transported in the time axis, and the strategic areas that describe the system to be analyzed in a complete and integrated manner.

#### 3.1.1. Key Future Drivers

The key future drivers can be described as guidelines that will modify the trend in the long run, and they will define the new future scenarios. We identified seven guidelines that are focused on providing approaches that will allow for framing the possible scenarios for the electricity sector in Mexico. These key future drivers are: Citizen participation, Social demand of energy, Corporate vision of profitability, Sustainability awareness, Governance, Technological management, and National promotion of R&D&I.

Citizen participation is the degree to which individuals become involved in the state and non-state public space, focusing on the identity of citizens as members of a political community. This participation is linked to the models of democracy and the type of government–society relationship that one wants to build, which will depend on whether citizens behave indifferently about the energy they consume or are interested in the origin and implications that their energy consumption entails or partners that generate, consume and actively manage their energy resources.

Social demand of energy does not refer to the number of energy units consumed in the country but to the satisfaction of energy needs, as well as respect for the uses and customs of the different sectors of society. This claim may simply be basic services that permit access to reliable and clean energy sources; intermediate services that allow them to promote potential community premises; or advanced services that develop the capabilities of individuals, organizations, and institutions.

Corporate vision of profitability reflects the aspiration to obtain the greatest amount of benefits from the fixed and variable investment costs involved in the operation, maintenance, administration and expansion of the National Electric System (SEN). The benefits are not only economic but also social and environmental. The vision of transferences is one in which subsidies continue to be provided from tax collection; that of solvency It is in which all the costs of the SEN are covered from the energy supply and purchase and sale contracts; and the vision of welfare is in which not only the economic costs and benefits are taken into account, but also the social, environmental and institutional ones.

Sustainability awareness is a process of continuous reflection on the management of the resources necessary to satisfy the needs of societies, both current and future. The most restricted consciousness is economic, which considers that maintaining growth ensures the well-being of society. On the other hand, environmental awareness also takes into account that natural resources are finite, so it is necessary to conserve and manage natural resources. Finally, the most developed degree of consciousness is expressed in the sustainable development goals (SDGs), which seek to overcome global challenges by taking into account the four axes of sustainability: economy, society, institutions and environment.

Governance is the art or way of governing that aims to achieve lasting sustainable development, promoting a healthy balance between the State, civil society and economic actors. This is dependent when decisions are strongly biased by foreign interests; is mixed when you have some influence over the possible lines of action; and is sovereign when the decision-making power lies entirely with national stakeholders.

The technological management of the Mexican power sector (SEN, due to its initials in Spanish, Sistema Eléctrico Nacional) consists of the control and operation of the different elements that make up the electrical infrastructure—that is, the generation plants, the transmission network, and the distribution networks. The management basic lies in centralized and large-scale decision-making, based on inflexible operating criteria. Smart management considers flexible alternatives to allow the dispatch of plants that generate electricity at a lower cost or with fewer polluting emissions. Digitalization allows the SEN to be measured and operated in real-time, given the dynamics introduced by the growing integration of distributed energy generation and storage technologies, as well as electric vehicles.

National promotion of R&D&I refers to the degree of economic and social support that the country's government provides to research, technological development and innovation. Marginal promotion allows the academic sector to survive with minimal resources; restricted promotion is when universities and research centers have resources to carry out their research up to low levels of technological maturity; and decided promotion is when the scientific community has a leading role in the proposal and generation of efficient dynamics for all sectors—economic, environmental and educational, in particular. This element will define the course of the country's development.

### 3.1.2. Strategic Areas

Strategic areas are considered those that are the fundamental areas of the SEN and that affect all the others, being able to describe in a systemic way the general behavior of the SEN. The eight strategic areas (SA) will allow us to know the bases for carrying out the prospective analysis. These areas are: Bioclimatic regionalization, Gender mainstreaming, Energy poverty, Financing, Human rights, Integrative infrastructure, Environment and Well-being.

Bioclimatic regionalization involves the division of a territory into smaller areas with common characteristics and represents a tool for the analysis of the geographical heterogeneities that exist in Mexico [37]. This area recognizes the geographical heterogeneities that exist in the SEN, which are related to the end uses of electricity; regional needs, potential and capabilities; climatic variables; the available natural resources; and the existing electrical and communications infrastructure. It is not considered from a local approach, but rather a regional one. Mexico is divided into 13 bioclimatic regions according to the "Modified Climate Classification System", which adjusted the Köppen classification system to the conditions of the Mexican Republic in 1964 and has been updated on four occasions by using the most recent data and increasing its resolution; the last update was in 2004 [38].

Gender mainstreaming is a fundamental strategic area in building the future and generating equality and equity between men and women through providing access to electricity for the entire population, preferably with the use of renewable energy. This aligns with one of the premises of the gender perspective, which is to improve the lives of people, societies, and countries so that the possibility of developing new content for socialization and relationships between human beings opens up [39]. This contributes to objectives 5 and 10 of sustainable development, which state gender equality and the reduction of inequalities.

Energy poverty is understood as the absence of options to access energy services in an adequate, affordable, reliable, high-quality, and safe way that is environmentally benign and contributes to economic and human development [40]. Because access to electricity is a basic service and many other things depend on it, such as refrigeration, heating, lighting, etc., it must be considered within prospective planning for 2050 in order to identify the change in energy poverty.

Financing will allow us to know if there are the necessary economic resources to encourage a flexible mechanism that allows electric energy to reach all Mexican economic sectors. In addition, social, institutional and environmental aspects must be considered. To

achieve this, we need to achieve the common good of electrification, preferably with the use of renewable energies in Mexico.

Human rights are well established within the Universal Declaration of Human Rights, issued by the United Nations Organization, considering the sustainable development objectives, which states that all human beings have the right to life, to be born free and equal and to have access to energy, among other things. For this strategic area, the objective is to provide a way for the entire population to have access to energy regardless of their race, sex, physical condition and social condition.

Integrative infrastructure refers to the components necessary to have a robust and resilient electrical network. In particular, to achieve the goal of electrifying the country, it is necessary to consider all the inputs for distributed generation, which can be tangible or intangible. In the case of the first, they are: the hardware, equipment and machinery required; and in the case of the second, they are: the software, the regulatory framework, the procedure manuals, etc. These will provide access to energy to all economic sectors of the country.

The environment and the power sector are closely related. On the one hand, all primary energy sources used for electricity generation come from the environment, whether fossil, renewable or nuclear. On the other hand, conventional electricity generation entails a series of environmental impacts, mostly negative, on continental and oceanic ecosystems throughout the different stages of the life cycle of the generation plants. In addition to this, the environment is the sociodemographic space where socioecological systems develop—that is, human beings and living entities interact and develop in a certain territory with the aim of continuing to preserve life on planet Earth.

Well-being is the state of the person whose physical and mental conditions provide a feeling of satisfaction and tranquility. Economic well-being is the state of the person whose economic conditions allow him to live in peace. Imagining the evolution of this strategic area will allow us to know if and how people in Mexican society are improving (or not) their level and quality of life. Among the 17 goals of sustainable development, number 3 refers to health and well-being.

## 4. Scenarios: Mexico in 2050

By carefully analyzing the strategic areas under the light of those key future drivers, we have designed four different futures for Mexico: BAU, Sovereignty, Market, and Sustainability.

### 4.1. BAU Scenario

The trend, portrayed by the BAU scenario (Business As Usual), describes a future where the policies prior to 2015 were kept and followed.

People, organizations and institutions are not willing to engage in dialog about the energy needs and capacities that exist in the region where they live. Energy companies offer electrical energy indistinctly in the various regions of the country. The domestic subsidy level is determined only based on local temperature. Energy generation depends mostly on private and foreign systems and products. The sources from which the energy comes are not a considered priority. There is no interest in R&D&I at the federal level, nor in the regions of the country. Support is minimal for the academic sector and the productive sector with national content. They continue to obtain technology from abroad.

Energy needs and capacities are considered homogeneous and do not depend on gender or sex, reinforcing existing inequalities and stereotypes by omission. Decisions are made seeking the greatest economic return. The value chain is integrated with a short-term vision, so the acquisition of technology, inputs and talent comes mainly from abroad. The electricity sector follows historical trends regarding the gender perspective. Women's and men's capacities and roles within the promotion of R&D&I according to their regions are not considered.

Energy poverty is not perceived as a problem in itself but as a component of poverty (in the form of lack of access to supply). Energy companies do not care about providing

affordable or reliable service. The focus is on providing access to electricity supply, which is subsidized by region; however, quality is not considered and neither is whether mitigation measures focus on discounts, subsidies and rates.

Electrical system operators do not facilitate the interconnection of distributed generation and storage technologies. Investment decisions in the sector remain in the hands of the State, which reduces consultations in the event of conflicts in the development of infrastructure projects. The private sector delegates communication interventions with citizens to the State. The State makes public investments, creating conditions for the economy to grow alongside the expansion of the sector. Investment in capital-intensive areas and the receipt of technological transfers is carried out through partnerships with private capital.

People ignore the possible benefits they can obtain and the capacity for active participation they can have. Energy companies do not make known the different sources of energy, and society comes to ignore the right of access to energy services and therefore cannot choose the source from which they come. Users are not interested in knowing other impacts besides the economic ones, and for this reason, the human rights (HRs) that will be violated are not considered when considering only the economic aspect; foreign interests have a greater impact. The fundamental focus of interest groups is to provide energy without considering the rights of the people and the ways in which that energy will arrive.

Electrical service users care only about having access to the network and being able to use electricity when they require it. Companies focus on providing access to electric service and provide maintenance to existing infrastructure in accordance with the growth of user demand. Electricity rates do not reflect the real cost of the infrastructure necessary to provide electricity services to the residential and agricultural sectors, which discourages energy saving, energy efficiency and distributed generation solutions. The Mexican electricity market and regulations are a "tropicalized" version of foreign regulatory frameworks. Energy companies prefer to hire foreign specialists. The electrical system is vertical.

The negative impact on the environment is only claimed when it negatively impacts the economic value of a space. Energy companies provide access to energy regardless of environmental damage. There is no interest in using non-conventional energies for energy generation. Resources for research are limited and focused. There is no promotion of R&D&I in environmental and social issues. People are interested in receiving the unit of energy at the lowest possible cost. The concept of sustainability has not been understood and economic sustainability is considered exclusively for development. Support for R&D&I is marginal.

### 4.2. Market Scenario

This scenario is designed by maintaining the policies put in place with Mexico's Energy Reform in 2015 that opened the electricity market and promoted renewable energies based on economic criteria. So, in 2050:

People, organizations and institutions discuss energy needs and regional capacities, although they do not commit to taking actions that require much time or money. Public and private bodies focused on productive activities receive electricity according to their specific and bioregional needs. Companies and civil society are interested in environmental and economic aspects. The needs, culture, uses and customs of the regions are not considered. The sources from which the energy comes are not a priority either. R&D&I promotion is provided only in sectors of interest to the public sector and few bioregions with specific political interests are considered. Furthermore, the guideline for the development of research is set by the government leadership with a short- and medium-term vision.

It is considered that energy needs and capacities are homogeneous and do not depend on sex or gender, reinforcing existing inequalities and stereotypes by omission. Decisions are made seeking the greatest economic return. The value chain is integrated with a short-term vision, so the acquisition of technology, inputs and talent comes mainly from abroad. Efforts to integrate greater generation capacity from renewable sources bring with

them a paradigm shift in the electricity sector, which includes significant participation of women in decision-making, operation, management, research and maintenance spaces in the electricity sector.

Energy poverty is not perceived as a problem in itself but as a component of multi-factorial poverty, and the structural causes of poverty are not recognized as an individual responsibility/failure. Companies comply with ensuring access to electricity supply, which is subsidized by region. Mitigation measures focus on discounts, subsidies and rates and there is a methodology to easily identify households in energy poverty. There is a national strategy but there is no entity in charge. Experts are involved in the design of public policies. Intelligent management of the national electrical system allows it to operate within the permitted limits, so the integration of distributed technologies is not limited. The local impact of the proposed public policies is analyzed.

The opening of the market system allows for a broader participation of actors in financial areas seeking to invest in the renewable energy sector. To this end, civil associations, cooperatives or SMEs in the energy sector are created that seek opportunities in new instruments to invest in energy projects and mechanisms for purchasing and selling technologies and/or electricity. The opening, liberalization and deregulation of the energy sector generates a condition of co-government between the State and the private sector. The development of renewable energy projects remains in the hands of transnational companies. The local capital would be an intermediary or secondary partner in some productive or service chain.

Companies in the electricity sector are required to consider the critical capacity and proposals of society, as well as take into account the right to information (how it works, publicize environmental and social impact studies) and mechanisms to follow up on the prevention of violations of the human rights of users or the population near energy generation sites. Human rights are not a priority for the use of natural resources.

Electric service users know the regulatory framework and structure of the electricity sector, at least in the regulation that applies to them. Companies focus on providing access to electrical services, providing maintenance to existing infrastructure, and expanding their capacity according to the growth of user demand, seeking to recover their economic costs through rates and contracts. The design of the Mexican electricity market and regulation is a "tropicalized" version of foreign regulatory frameworks, and energy companies prefer to hire foreign specialists. The transmission network is strengthened to be able to transport energy from where the cheapest and best quality energy resource is located to the consumption centers. Meteorological data and predictive algorithms are considered for the operation of the system. There is no link between academia, industry and government to scale developments.

The negative impact on the environment is only claimed when it negatively impacts the economic value of a space. Access to energy is provided reliably and the use of renewable energy and care for the environment are prioritized. Energy generation with renewable sources is considered mainly for economic benefit and there is little consideration towards socio-ecological systems. Non-conventional sources are considered to generate electricity at a lower economic and environmental cost. There is minimal digitization of the sector. Resources for research are limited and focused, including environmental and social issues.

Civil society only understands the use of energy as related to economic well-being. People are interested in receiving the unit of energy at the lowest possible cost. The economic well-being of people resides exclusively in the public policies issued by the federal, state and municipal governments. The SEN operates to cover the basic services of people, communities and companies that promote economic development. Research, development and innovation are focused on satisfying the social well-being of the population in a restricted way.

*4.3. Sovereignty Scenario*

This scenario follows from the application of the policy framework depicted in the "Estrategia de Transición para Promover el Uso de Tecnologías y Combustibles más Limpios, en términos de la Ley de Transición Energética" published in February 2020 [41].

People, organizations and institutions are not willing to engage in dialog about the energy needs and capacities that exist in the region where they live. Public and private bodies focused on productive activities receive electricity according to their specific and bioregional needs. Energy companies charge subsidized rates to the domestic sector, based on local temperature. PPAs are established in the institutional area in strategic projects to resolve the needs of the population in each bioregion. The needs, culture, uses and customs of the regions are not considered. The sources from which the energy comes are not a priority either. The guidelines for the development of research are set by the government leadership, with a short-term vision.

Information is generated on energy consumption and the energy sector is disaggregated by sex, and from that understanding, efforts are made not to reinforce gender inequalities and stereotypes. The social demand for energy is not disaggregated by gender. The heterogeneities of energy needs between men and women according to the region where they live are not recognized. Decisions are made seeking the greatest economic return. The CFE considers that the gender perspective is a means to achieve development goals, taking into account the norms, roles and access to resources of people to train talent. The industry and the community leaders are mostly men.

Energy poverty is recognized as a condition that goes beyond access to supply; energy efficiency solutions, access to distributed generation and storage systems, and differentiated rates by region need to be implemented. There are mechanisms to defend people, programs to explain energy consumption, and associations dedicated to support. There are no national legal frameworks to mitigate and address energy poverty. Electrical system operators do not facilitate the interconnection of distributed generation and storage technologies. A constant observation of energy poverty is carried out at different levels from an interdisciplinary perspective. The government sector collaborates strongly with academic institutions on the design, implementation and evaluation of public policies and programs aimed at mitigating and eradicating energy poverty in Mexico.

The State resumes the processes of financial planning and stability of energy parastatals in the long term, considering energy as an essential service and right of the population in the different regions of the country. The development of energy projects is driven by public financing, where private capital supports areas of high added value, such as the acquisition and installation of technology and infrastructure that the country does not develop endogenously. Public finances have the objective of supporting parastatal companies, replacing the largest amount of installed capacity that comes from abroad, and come from internal public savings, sometimes allying with private capital or external financing channels (governmental from other countries, corporate, stock exchange and banking).

People ignore the benefits they can obtain and the capacity for active participation they can have. Energy companies do not make known the different sources of energy, and society comes to ignore the right of access to energy services and therefore cannot choose the source from which they come. In the absence of applying monitoring mechanisms for compliance with human rights, these can become violated. Companies do not apply social responsibility. Users are not interested in knowing other impacts besides the economic ones, and for this reason, the DHs that will be violated are not considered when considering only the economic aspect. Public institutions and civil society work together through consultations with indigenous peoples and communities on the impacts and monitoring of respect for human rights.

Users of the electrical service care about having access to the network and being able to use electricity when they require it. Companies focus on providing access to electrical services, providing maintenance to existing infrastructure, and expanding their capacity

according to the growth of user demand. Electricity rates do not reflect the real cost of the infrastructure necessary to provide electricity services to the residential and agricultural sectors, which discourages energy saving, energy efficiency and distributed generation solutions. Some technology companies design and manufacture their equipment in Mexico. An electricity market and a regulatory framework are designed according to the needs of Mexico. The investment capital for the construction of tangible infrastructure comes from both the national public sector and national and international private sectors. The electrical system is vertical.

Users demand that companies in the electricity sector take environmental protection measures. There is interest on the part of energy companies to provide access to energy reliably, regardless of environmental damage. Energy generation with renewable sources is considered mainly for economic benefit and there is little consideration towards socio-ecological systems. Sustainability is seen from an economic perspective beyond the needs of society. The energy systems that are acquired do not consider environmental issues among their criteria. There are no environmental considerations in management. Basic and applied research is carried out on environmental issues, but it is not linked to strategic energy projects.

People are interested in receiving the unit of energy at the lowest possible cost. Corporations only consider the economic well-being of users. The concept of sustainability has not been understood, and economic sustainability is considered exclusively for development. The economic well-being of people resides exclusively in the public policies issued by the federal, state and municipal governments. The SEN operates to cover the basic services of people, communities and companies that promote economic development. Research, development and innovation are focused on satisfying the social well-being of the population in a restricted way.

*4.4. Sustainability Scenario*

This scenario describes a country where sustainability is mainstream, and there is a conviction that a deep commitment to a different way of living, where community well-being is in harmony with its surroundings, is central to any decision-making process.

People, organizations and institutions discuss the energy needs and capacities that exist in the region where they live. Public and private bodies focused on productive activities receive electricity according to their specific and bioregional needs. Energy companies charge rates without subsidies. The rates reflect the different regional costs, achieving their economic viability. Civil society, government entities and companies align to develop policies that allow social needs to be resolved in a comprehensive manner by simultaneously considering the environmental, economic, social and institutional impacts, considering the characteristics of each bioregion. Renewable energies are used to satisfy all demand, and there is complete energy sovereignty in the country. There is an infrastructure to generate, transmit, distribute and consume energy in a permanent relationship with the needs of energy companies, civil society and the government, according to the characteristics of each bioregion. The guideline for R&D&I is set by the government leadership.

Information is generated on energy consumption and the energy sector is disaggregated by sex, and from that understanding, efforts are made not to reinforce gender inequalities and stereotypes. There is an approach where the authority weighs the energy production needs, considering the norms, roles and access to resources of each gender, to achieve the development goals set out in the National and Regional Plans. The CFE considers that the gender perspective is a means to achieve development goals, taking into account the norms, roles and access to resources of people to train talent. Efforts to integrate greater generation capacity from renewable sources bring with them a paradigm shift in the electricity sector, which includes significant participation of women in decision-making, operation, management, research and maintenance spaces of the electrical sector. Greater inclusive participation with a gender perspective is promoted to meet the goals of the R&D&I development plan at the national and regional levels.

Energy poverty is recognized as a condition that goes beyond access to supply and it is understood in terms of satisfaction of energy needs. There are programs to explain energy consumption and associations dedicated to support. It is still understood in terms of the user, and the individual social factors of each person. Energy companies ensure a reliable and quality service for users. They understand that it is not enough to provide access to the electrical grid, but that energy efficiency solutions, access to distributed generation and storage systems, and differentiated rates by region need to be implemented. The elimination of subsidies increases the possibility of using the saved resources to promote new energy policies that help mitigate the negative effects of the elimination of subsidies. The measures for its mitigation are regionalized, take into account socio-cultural aspects, and are, at a minimum, gender-sensitive. Energy poverty is considered a key variable in the country's sustainable development. There is a national strategy with an entity in charge. The intelligent management of the national electrical system allows it to operate within the permitted limits, so the integration of distributed technologies is not limited. People's energy consumption is monitored, and a catalog of consumption profiles is built from the standard.

The financial sector is open to environmental activism that allows for the participation of diverse actors committed to sustainability. In addition to traditional sources of financing (corporate, state and mixed), alternative, hybrid and social sources are added that address a plurality of environmental problems in various regions. This implements a series of participatory institutions (civil associations, cooperatives, etc.) that capture and channel financing towards energy projects, whether distributed or interconnected, where citizens are part of the investment decision-making processes. The concept of "cost" ceases to be an exclusively monetary issue and is related to social and environmental costs. Profitability also becomes an indicator of economic, social and public well-being.

People and companies in the electricity sector implement prevention and mitigation activities to guarantee that the human rights of people living in the same area of influence are not violated. Energy companies come to consider the energy needs that users and the community want to cover, as well as the way in which energy can be produced without leaving aside the benefit of both parties and guaranteeing the right to a clean environment. The social and environmental benefits are taken into account apart from the economic ones; the population is aware of them and the benefit for both parties is considered, guaranteeing each of the HRs. Social responsibility is taken into account. Public institutions and civil society work together through consultations with indigenous peoples and communities on the impacts and monitoring of respect for human rights. There are guidelines regarding human rights that focus on the lines of action for the academic, private and public sectors.

Electrical service users analyze the integration of electricity generation, storage and management technologies in their properties. Energy companies recognize community needs and expand their infrastructure so that they can exploit their productive potential. Energy companies recover their economic costs through rates and contracts. Comprehensive planning is carried out in the short, medium and long term with the aim of achieving access to reliable, affordable, clean and good-quality energy sources for all people. The investment capital comes from both the national public sector and national and international private sectors. The transmission network is strengthened to be able to transport energy from where the cheapest and best quality energy resource is located to the consumption centers, with the expeditious integration of distributed generation and storage technologies. There is close collaboration between academia, society, government and industry in designing, implementing and evaluating solutions to enable the large-scale integration of distributed generation and storage systems.

Users take advantage of the renewable resources available in a sustainable manner to satisfy their energy needs. Access to energy is provided reliably and the use of renewable energies is prioritized, and synergy between the environment and the productive capacity of the regions and their natural resources. Social, environmental, institutional and economic finance profitability is considered. Regulatory entities are committed to working for the

benefit of the environment and with a global approach with local actions. Non-conventional sources generate electricity at a lower economic and environmental cost. Basic and applied research is carried out on multisectoral issues, linked to strategic energy projects.

Society is supplied with electricity to satisfy health and educational needs, as well as economic and environmental needs, at a price accessible to everyone. Corporations have proven social responsibility through their actions in the economic, health, educational and environmental fields, without harming the environment and social well-being; this has been achieved in accordance with the resilience of particular ecosystems. The social well-being of the population, both in health and education, as well as in the economy and environment, is achieved when governments and civil society jointly agree on development plans. Society and companies have agreed to carry out proper SEN management. Research, development, and innovation are focused on satisfying the social well-being of the population in an unrestricted way.

## 5. Conclusions and Policy Implications

In this work, a prospective analysis of the possible $CO_2$e emissions has been conducted according to different possible energy policies in Mexico in 2050 with the LEAP scenario modeling software, carrying out an economic optimization that accounts for the lowest emissions possible as a second optimization criterion. It was shown that the National Energy Programs of the two Mexican federal administrations from the previous administration of 2012–2018 and the current one of 2018–2024 (represented by the Market and Sovereignty scenarios, respectively) had no proper strategies to achieve the agreements adopted by Mexico in the Paris Agreement. In 2015, these emissions were 150 Mt of $CO_2$e, and the Mexican goal for 2050 is 50. According to this study, the estimated results for each scenario related to the goal are: BAU and Sovereign: 4.6 times; Market: 3.4; and Sustainable: 1.

It is only when the use of the possible renewable energy resources in Mexico, represented in the Sustainable scenario, are considered that the goals are achieved. This means that the use of renewable energies should not only be promoted from the installation due to the viable technical and economic issues, but feasibility studies must also consider environmental and social impacts (which must be considered in further investigations related to the implementation of future installations) to turn these possible resources into proven ones. In addition, it is not possible from the point of sovereignty to depend on huge volumes of imported natural gas from now to 2050.

Complementing the mathematical modeling, we applied a proprietary methodology based on Foresight Frameworks. Multidisciplinary analysis and discussions were undertaken to depict the different future scenarios that account for those four different Mexico designs.

In order to fully achieve the country's NDCs by 2050, according to the Sustainable scenario, the vision of a Mexican Energy Plan should consider: a deep commitment to a different way of living, where community well-being in harmony with its surroundings is central to any decision-making process. People, organizations, and institutions agree on energy needs and capacities without gender biases and guaranteeing human rights. Energy consumption information is generated and disseminated. Energy poverty is recognized as a condition that goes beyond access to supply. The financial sector is open to environmental activism that allows for the participation of diverse actors committed to sustainability, with alternative, hybrid and social sources added. Electrical service users analyze the integration of electricity generation, storage and management technologies and take advantage of the renewable resources available in a sustainable manner to satisfy their energy needs. And finally, society is willing to know the impacts on health and education, in addition to the economic and environmental impacts, of energy supply projects.

This study shows that the best way to face the future and the coming crisis is with a power sector that takes advantage of the different renewable energy resources of the country and presents an energy-independent option that is strengthened through the diversity of energy sources, thus moving toward achieving a sustainable future and complying with

our approved NDCs. However, to make this possible, policies should be designed to foster the society, government and financial ecosystem that aligns with the SDGs in a holistic comprehensive manner.

**Author Contributions:** Conceptualization, M.M. and K.G.C.-V.; methodology, M.M.; software, J.C.C.; validation, M.M. and J.C.C.; formal analysis, and investigation M.M., J.C.C., S.Á.-H. and K.G.C.-V.; resources, M.M. and K.G.C.-V.; data curation, J.C.C.; writing—original draft preparation, M.M., J.C.C., S.Á.-H. and K.G.C.-V.; writing—review and editing, M.M., K.G.C.-V., J.C.C. and C.D.L.-F.; visualization, J.C.C.; supervision, M.M. and C.D.L.-F.; project administration, K.G.C.-V.; funding acquisition, M.M. All authors have read and agreed to the published version of the manuscript.

**Funding:** This research and the APC was funded by UNAM, through the project titled "Validation of Methodology for Integration and Updating of Energy Scenarios for Mexico by 2050", grant number PAPIIT IN111623.

**Institutional Review Board Statement:** Not applicable.

**Informed Consent Statement:** Not applicable.

**Data Availability Statement:** All data used in this study is publicly available. The authors will provide it by request.

**Acknowledgments:** To the Stockholm Environment Institute for providing academic licenses to access and utilize the Low Emissions Analysis Platform (LEAP) software. To Diocelina Toledo-Vázquez, Leonardo Bañuelos-Ruiz, Cesia de Paz-Bautista, Araceli Granados-Hernández, Dulce Becerra-Paniagua, Alberto Avila-Nuñez, Karla Ricalde and Harriet Thomson for the support given in the realization of this study.

**Conflicts of Interest:** The authors declare no conflict of interest. The funders had no role in the design of the study; in the collection, analyses, or interpretation of data; in the writing of the manuscript; or in the decision to publish the results.

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
