# Peer review of "Paving Paths to 2050: Mapping the Mexican Power Sector’s Potential to Build Sustainable Futures"

_sustainability, doi:10.3390/su16010068_

Round 1
Reviewer 1 Report
Comments and Suggestions for Authors
Comments
In this work, a prospective analysis of the possible CO2e emissions has been made, according to different possible energy policies in Mexico by 2050, with the LEAP scenario modelling software, making an economic optimization accounting for the lowest emissions possible as second optimization criteria. The topic of this research is interesting. I would suggest the publication of this article if the following minor issues are addressed.
- In the abstract, the explanation of the methodology is concise and clear. However, it might be helpful to briefly explain what LEAP scenario modelling is, as not all readers may be familiar with this tool.
- There are a few minor grammatical issues that can be addressed, such as the use of "a the" in sentence 15 of the abstract.
- Some acronyms are introduced without being defined first (e.g., PRODESEN, ENCC). It would be useful to provide the full name followed by the acronym in parentheses upon first mention.
- The results and conclusion are clearly stated. However, consider elaborating on the implications of the Sustainable scenario for Mexico's energy policy and how it could contribute to meeting the country's NDCs (Nationally Determined Contributions) under the Paris Agreement.
- In the methodology, the references to specific reports and studies provide a solid basis for the methodology. However, it is important to ensure that all referenced materials are accurately cited and formatted according to the publication's style guidelines.
- Based on the LEAP model results, the authors have identified the dominant energy sources in each scenario and the implications for CO2 emissions and dependency on specific energy sources.
- The conclusion provides a clear and concise summary of the study's findings and its implications for Mexico's energy policy. However, it could be enhanced with a few improvements:
o Clarity and coherence: The conclusion should be clearer and more coherent in presenting the main findings of the study. For example, the statement "neither the previous administration or the current one, represented by the Market and Sovereignty scenarios, have planned, and done is sufficient to comply with the agreements adopted by Mexico in the Paris Agreement" could be simplified to enhance readability.
o Quantitative data: Including quantitative data on the potential CO2e emissions reductions from adopting renewable energy in Mexico would strengthen the argument.
o The conclusion could also suggest areas for future research, such as investigating the social and environmental impacts of renewable energy installations in Mexico.
Comments on the Quality of English Language
There are a few minor grammatical issues that should be addressed
Author Response
In this work, a prospective analysis of the possible CO2e emissions has been made, according to different possible energy policies in Mexico by 2050, with the LEAP scenario modelling software, making an economic optimization accounting for the lowest emissions possible as second optimization criteria. The topic of this research is interesting. I would suggest the publication of this article if the following minor issues are addressed.
Thank you very much for your comments, suggestions and considerations to our research. We have integrated all of them. We are indicating the line number where the corresponding correction can be found.
1 In the abstract, the explanation of the methodology is concise and clear. However, it might be helpful to briefly explain what LEAP scenario modelling is, as not all readers may be familiar with this tool.
Done, line 12.
2 There are a few minor grammatical issues that can be addressed, such as the use of "a the" in sentence 15 of the abstract.
Corrected, thank you.
3 Some acronyms are introduced without being defined first (e.g., PRODESEN, ENCC). It would be useful to provide the full name followed by the acronym in parentheses upon first mention.
Done, lines 191 and 87.
4 The results and conclusion are clearly stated. However, consider elaborating on the implications of the Sustainable scenario for Mexico's energy policy and how it could contribute to meeting the country's NDCs (Nationally Determined Contributions) under the Paris Agreement.
Thank you, we have done it, lines from 1082 to 1095.
5 In the methodology, the references to specific reports and studies provide a solid basis for the methodology. However, it is important to ensure that all referenced materials are accurately cited and formatted according to the publication's style guidelines.
We have done it, thank you.
6 Based on the LEAP model results, the authors have identified the dominant energy sources in each scenario and the implications for CO2 emissions and dependency on specific energy sources.
7 The conclusion provides a clear and concise summary of the study's findings and its implications for Mexico's energy policy. However, it could be enhanced with a few improvements:
8 Clarity and coherence: The conclusion should be clearer and more coherent in presenting the main findings of the study. For example, the statement "neither the previous administration or the current one, represented by the Market and Sovereignty scenarios, have planned, and done is sufficient to comply with the agreements adopted by Mexico in the Paris Agreement" could be simplified to enhance readability.
Thanks, we have done it, lines 1066 to 1070.
9 Quantitative data: Including quantitative data on the potential CO2e emissions reductions from adopting renewable energy in Mexico would strengthen the argument.
We did it, line 1066 to 1068.
10 The conclusion could also suggest areas for future research, such as investigating the social and environmental impacts of renewable energy installations in Mexico.
We included it, line 1074.
Reviewer 2 Report
Comments and Suggestions for Authors
The authors regarding the possibilities that the Mexican Power Sector has to comply with the commitments adopted with the international community towards the mitigation of climate change.
1. In the abstract, many first appearance abbreviations without full names given.
2. Please explain the source of the power sector data and the future data forecast.
3. How is the relationship between power sector analysis and CO2 emissions (line 1023) established?
Comments on the Quality of English LanguageMinor revision.
Author Response
Reviewer 2
The authors regarding the possibilities that the Mexican Power Sector has to comply with the commitments adopted with the international community towards the mitigation of climate change.
Thank you very much for your suggestions and comments. We answer each one in the following lines.
- In the abstract, many first appearance abbreviations without full names given.
Thank you, we have corrected them.
- Please explain the source of the power sector data and the future data forecast. We have explained them more clearly, thanks for pointing this out. It can be found in lines: 98 INEGYCEI, 154 CFE, 187 PRODESEN, 203 IEA, 204 SENER and CONAPO, 215 SENER, 221 IRENA, 222 NREL, 224 COPAR, and 231 ENCC,
- How is the relationship between power sector analysis and CO2 emissions (line 1023) established? Thank you, around line 1023, in the context of explaining the Sustainable Scenario characteristics, it was written that… “Expenses and costs as well as care for the environment at all times are collaborating transversally and in alignment with the SDGs. Profitability is more than economic, since social, environmental, institutional and eco-social finance profitability is also considered. Regulatory entities are committed to working for the benefit of the environment and with a global approach with local actions.” This is to emphasize that economic profitability alone has not been sufficient to improve the well-being of the majority of the population; one most considered the other pillars of sustainable development: social, environmental, and institutional.
Reviewer 3 Report
Comments and Suggestions for Authors
1. It is desirable to reconstruct the article's abstract according to the structure adopted in the scientific journals (relevance of the problem, purpose, methodology, and results).
2. Кeywords donʼt reflect the scientific content of the study.
It is desirable to emphasise which scientific problem the study's authors will solve. What scientific gap exists concerning its solution, and how does the authors' proposed methodology solve it?
3. It is desirable to determine the purpose of the research at the beginning of the article.
4. Authors must rebuild the article's structure following the accepted practice of scientific journals: introduction, literature review, methodology, results, etc.
5. The authors should clearly emphasise the advantages of the proposed research methodology.
6. In general, all paper figures are difficult to read.
7. Lines 256-259 look like comments to the article.
8. Chapter 4, which characterises the scenarios, should be shortened, and specific measures within each scenario should be defined. It should contain specific management recommendations.
9. There is no need to define commonly known categories such as energy poverty, Financing, Human rights, Environment, well-being, etc.
10. In general, the wording of the article should be more scientific.
Author Response
Reviewer 3
- It is desirable to reconstruct the article's abstract according to the structure adopted in the scientific journals (relevance of the problem, purpose, methodology, and results).
Thank you for your observation but we considered that It was not desirable, starting the abstract, to mention that no country, including Mexico, will be able to achieve the commitments made in the Paris Agreement.
- Кeywords donʼt reflect the scientific content of the study.
We reviewed the keywords and corrected a couple of them. thank you for pointing this out.
It is desirable to emphasise which scientific problem the study's authors will solve.
In line 22 the research question is presented.
What scientific gap exists concerning its solution
We consider that it is stated in lines 105 to 111,
and how does the authors' proposed methodology solve it?
We elaborated it more precisely in the conclusions.
- It is desirable to determine the purpose of the research at the beginning of the article.
It can be found in Lines 9 to 12 and 105 to 111.
- Authors must rebuild the article's structure following the accepted practice of scientific journals: introduction, literature review, methodology, results, etc.
Except for methodology and results this structure was followed. This structure was used because he apply two different methods for the analysis. In this way, the assumptions and results are clearer by keeping them together for each approach.
- The authors should clearly emphasise the advantages of the proposed research methodology.
Thank you, we have corrected this, lines 108 to 118.
- In general, all paper figures are difficult to read.The quality of the figures have been improved, thank you.
- Lines 256-259 look like comments to the article. Sorry, we couldn’t find the comments you mention. Those lines correspond to the descriptions of the four scenarios considered
- Chapter 4, which characterises the scenarios, should be shortened, and specific measures within each scenario should be defined. It should contain specific management recommendations. It has already been reduced, thank you for the suggestion.
- There is no need to define commonly known categories such as energy poverty, Financing, Human rights, Environment, well-being, etc. Please, indulge this apparent excess but we consider it is important to define concepts when working with persons of different disciplines. We have found in multidisciplinary studies that defining the concepts used is necessary to avoid confusion with other terms in different fields..
10. In general, the wording of the article should be more scientific. We have used a language that can be easily understood by different disciplines. As in social sciences, narrative scenarios with “images” that can be understood by persons of any discipline are presented. We hope you find that this explanation clarifies the type of language and approach we have used, as a multidisciplinary team.
Reviewer 4 Report
Comments and Suggestions for Authors
In this paper a prospective analysis of the Mexican Power system has been carried out; the authors also show the possibilities for complying with the commitments adopted at the Paris Agreement in the National Climate Change Strategy of the General Law for Climate Change.
The paper is very interesting; different scenarios are proposed and analysed with a clear methodology even if some improvements are necessary for a better comprehension and readability.
Some remarks:
Concerning the "Sustainable scenario": from the figure 5, we can see that wind energy receives an important improvement between 2036 and 2040. Then the wind energy production seems to be maintained constant even if the authors say "Mexico was the fourth country in Latin America with the highest wind potential" (page 4, line 133) and it seems to be understood that this type of energy can be developed in a more important way.
Perhaps some considerations about should be proposed, i.e. what kind of limitations are present?
Second remark (always for "Sustainable scenario", page 8): some indications about the ".. the incorporation of electricity storage systems..." should be proposed. These devices are fundamental for a correct electrical network functioning but its are economically important. Moreover, please, consider that: "Electrical system operators do not facilitate the interconnection of distributed generation and storage technologies.." (page 18, line 787-788) and maintenance problems for a new kind of technology could be consistent.
Finally, in my opinion paragraph 4 (Scenarios: Mexico in 2050) should be profoundly revised and made a little more streamlined (11 pages!) to allow for greater understanding of the salient points and the reading should be aided with diagrams or tables .
Please, modify:
page 5, line 214: "y" --> "and"
Page 10, line 383: "from sor identity" --> "from identity"
page 11, line 407: "... environmental awareness It also..." --> "...environmental awareness it also..."
Author Response
Reviewer 4
In this paper a prospective analysis of the Mexican Power system has been carried out; the authors also show the possibilities for complying with the commitments adopted at the Paris Agreement in the National Climate Change Strategy of the General Law for Climate Change.
The paper is very interesting; different scenarios are proposed and analysed with a clear methodology even if some improvements are necessary for a better comprehension and readability.
Thank you very much for your comments and suggestions, we have included them all. In the following lines you will find our specific answers.
Some remarks:
1 Concerning the "Sustainable scenario": from the figure 5, we can see that wind energy receives an important improvement between 2036 and 2040. Then the wind energy production seems to be maintained constant even if the authors say "Mexico was the fourth country in Latin America with the highest wind potential" (page 4, line 133) and it seems to be understood that this type of energy can be developed in a more important way.
Perhaps some considerations about should be proposed, i.e. what kind of limitations are present? Thank you, we have corrected it in lines 314 to 317.
2 Second remark (always for "Sustainable scenario", page 8): some indications about the ".. the incorporation of electricity storage systems..." should be proposed. These devices are fundamental for a correct electrical network functioning but its are economically important. Moreover, please, consider that: "Electrical system operators do not facilitate the interconnection of distributed generation and storage technologies.." (page 18, line 787-788) and maintenance problems for a new kind of technology could be consistent.
Than you we have corrected this also. Lines 313 and 314.
3 Finally, in my opinion paragraph 4 (Scenarios: Mexico in 2050) should be profoundly revised and made a little more streamlined (11 pages!) to allow for greater understanding of the salient points and the reading should be aided with diagrams or tables . Thanks we made an effort to reduced it as much as possible.I hope you find it more readable.
4 Please, modify:
page 5, line 214: "y" --> "and" Corrected. Line 219.
Page 10, line 383: "from sor identity" --> "from the identity" Corrected. Line 404
page 11, line 407: "... environmental awareness It also..." --> "...environmental awareness it also..." Corrected, line 429
Round 2
Reviewer 3 Report
Comments and Suggestions for Authors
No suggestion
Reviewer 4 Report
Comments and Suggestions for Authors
no further comments. I'm satisfied of the authors answers